# “Slowly but Steadily, You’re Running Out of Steam”: Aging Parents’ Caregiving Experiences Through Photovoice

**DOI:** 10.3390/ijerph22081297

**Published:** 2025-08-19

**Authors:** Martin Nagl-Cupal, Marlene Werner, Daniela Haselmayer, Thomas Falkenstein

**Affiliations:** 1Department of Nursing Science, Faculty of Social Sciences, University of Vienna, 1080 Vienna, Austria; marlene.werner@meduniwien.ac.at (M.W.); daniela.haselmayer@akhwien.at (D.H.); thomas.falkenstein@univie.ac.at (T.F.); 2Endowed Professorship in Nursing Science, Department of Primary Care Medicine, Center of Public Health, Medical University of Vienna, 1090 Vienna, Austria

**Keywords:** older adults, family caregivers, qualitative research, photovoice, participatory research, family health, persons with disabilities

## Abstract

Aging parents caring for adult children with disabilities or other care needs face significant challenges and health-related issues over extended periods. The aim of this study was to explore the experiences of aging parents with caregiving responsibilities. This qualitative participatory study followed the research process of a photovoice study. Thirteen parents, aged 51 to 76 years, of adult children with care needs, aged 20 to 49 years, participated in this study. The parents captured photographs depicting their daily lives which were contextualized and coded during group discussions. The data were analyzed using thematic analysis and reviewed collaboratively with the participants. Six key themes emerged in the data analysis. 1. mastering complexity, 2. being an expert and advocate, 3. balancing autonomy and care, 4. care as a lifelong journey, 5. standing on the margins of society, and 6. worrying about the future. This study underscores the urgent need for comprehensive and coordinated support systems that ensure the well-being of aging caregivers while addressing the evolving needs of their adult children.

## 1. Introduction

Family caregivers play a central role in providing care for individuals with care needs at home. As the population ages, the demand for informal care within families continues to grow, placing significant strain on caregivers’ physical, mental, and financial resources [1,2]. Over the past few decades, life expectancy has increased across various populations, including individuals with care needs and intellectual disabilities [3]. As a result, the duration of caregiving has also lengthened, often spanning the entire lifespan.

The public health sector and individuals with disabilities increasingly rely on unpaid family caregivers to provide the majority of home and community-based care [4]. This reliance presents multiple public health challenges and underscores the urgent need for comprehensive strategies to mitigate the negative impacts on the health and well-being of family caregivers.

Parents are often the primary caregivers for children with care needs, and many continue to provide care as they themselves age [5]. While there is no specific age at which caregivers are classified as “aging parents,” a study conducted in Austria found that 12% of family caregivers were parents [6]. Based on the age of their children, it is estimated that approximately 2–3% of all family caregivers provide home care for their adult children aged 19 to 40. Because this group represents a relatively small proportion of family caregivers, it is often overlooked.

Aging parents who care for their adult children with care needs are particularly vulnerable to social and economic deprivation due to their dual roles as caregivers and older adults. This responsibility negatively impacts their employment and financial stability [7,8], as well as their emotional and physical wellbeing [9], and significantly increases their risk of physical and mental health problems [8,10,11,12].

One of the most pressing concerns for these caregivers is uncertainty about the future [9]. While most prefer to continue providing care at home, many also view assisted living facilities as an acceptable option. However, suitable support structures to accommodate these preferences are often lacking [13,14]. Although most parents do not want their child to move to a traditional care facility, they recognize that this may become necessary if their child requires a level of care they can no longer provide [10].

In this study, we explored the experiences and challenges of aging parents who care for an adult child with care needs using the photovoice technique. This participatory approach aims to provide insight into this often overlooked group, empower participants through active engagement in the research process, and foster collective exchange among caregivers. This study is guided by the following research question: What are the experiences and challenges of aging parents with caregiving responsibilities?

## 2. Materials and Methods

Photovoice is a qualitative participatory research method in which participants use photography to document and share their experiences. It is shaped by critical consciousness, feminist theory, and nontraditional approaches to documentary photography [15]. Originally developed in community-based health research, photovoice aims to amplify the voices of underrepresented and marginalized groups [16], which also includes people with intellectual disabilities [17]. This method enables participants to identify and reflect on their community’s strengths and challenges, foster critical dialogue on relevant issues, and use visual data to engage policymakers [18]. The primary goals of a photovoice study include in-depth exploration, increased visibility, and participant empowerment, with photographs serving as catalysts for narratives and discussions within the community [18], which aligns with the research question of this study. The phases of the research process are outlined below [19].

1. Planning, Preparation, and Recruitment: Included were parents of adult children in need of care who wanted to talk about their situation and were willing to take photos of their situation and share them in a group discussion. To describe the project and recruit participants, we developed informational materials, including an accessible video and a website. Recruitment was conducted through self-help groups, social media, the Austrian Interest Group of Family Caregivers, and organizations supporting carers or people with disabilities, such as daycare centers and workshops. Twenty parents expressed interest in participating in this study. However, due to time constraints or illness, only 13 parents ultimately participated.

2. Data collection: Participants received training on procedures, technical aspects, ethical principles, and data protection during two online meetings [19]. During a two-week photo phase, parents captured 10–15 photographs of their caregiving responsibilities using smartphones or disposable cameras. To guide participants and clarify the project objectives, we provided stimulating statements [18] as shown here:

Stimulating statements for the photo phase:Take photos that show what your life looks like.Take photos that show what you are most concerned about in relation to caregiving—now and in the future.Take photos that show who or what helps you most to maintain care.Take photos that show what you would change immediately if you had “superhuman powers”.

3. Data analysis: To minimize the time commitment for participants, we conducted three video group discussions instead of face-to-face meetings, with each participant attending one session. A total of 158 photographs were submitted for discussion. The discussion was conducted in real time using a video conferencing tool. First, the participants agreed on the agenda. Afterwards, the participants showed their photos, which were then discussed with the other participants. The researchers were also free to ask questions. To support group discussions of photographs, we used the SHOWeD guide (Table 1) [20,21] which provides a dialogic structure to uncover the meanings behind the images and serves as a structured prompt to guide the analytical process. Although participants were informed that they were only expected to share the photos they felt comfortable showing, they ended up sharing and discussing all of their photos. A very trusting atmosphere in the three groups encouraged the other participants to share their own stories inspired by the photos of their group members.

The group discussions were recorded, transcribed verbatim, and analyzed using thematic analysis following the steps outlined by Braun and Clarke [22]. 1. *Familiarization with the data:* Transcripts of the three group discussions (7.5 h audio data) were reviewed to gain an in-depth understanding and identify patterns of interpretation. 2. *Initial coding:* Meaningful statements were systematically coded. 3. *Theme creation*: Themes were developed to provide an overarching structure, as illustrated in the thematic map. 4. *Theme refinement:* The structure was refined by reviewing and adjusting the themes. 5. *Theme definition and labeling:* Each theme was carefully defined and labeled. An example of the thematic analysis process is provided in Appendix A (see Figure A1). The empirical process of data collection and analysis took place between January and November 2022.

4. Ethics: Participants were provided with both verbal and written information about the research project, including its aims, procedures, and the ethical considerations involved in taking photographs of sensitive subjects. The selection and publication of photographs were carefully managed by the researchers, who held a special responsibility to protect participants’ rights and well-being [23].

After the results were finalized, additional consent was obtained for the use of selected images showing identifiable faces in publications. In cases where explicit consent from individuals or their legal representatives was not provided, images were pixelated to protect the identities of those depicted. Photographs that could compromise privacy, such as those showing car license plates, were altered to render identifying details unrecognizable. For reporting purposes, the names and personal data of the participants were pseudonymized to provide additional protection. For ethical reasons, all faces in the photographs were pixelated to protect the participants’ identities in the publication.

This study received prior approval from the Ethics Committee of the University of Vienna, ref. no. 00721] before data collection began.

5. Trustworthiness: A collaborative research approach was employed to enhance the confirmability of the data and ensure alignment between the researchers’ interpretations. Regular team meetings were held to discuss the analysis, coding, and results [24]. The use of photovoice, combined with the researchers’ prior experience working with family caregivers, fostered openness among participants, contributing to the collection of high-quality data.

To further ensure the validity of the findings, an additional participatory meeting was conducted with participants to collaboratively review and confirm the results. The following questions were asked in the group for each of the topics developed: What is easy to understand? What requires further explanation? Are there any specific recommendations for action you would like to share with us? The participants took this process very seriously and provided important input, which significantly enriched the findings. Their active engagement contributed to valuable and collaborative results.

## 3. Results

### 3.1. Description of Parents with Caring Responsibilities

The study sample included 13 parents from 13 families with caregiving responsibilities, consisting of 11 women and 2 men, aged 51 to 76 years (mean age = 58). Their adult children, for whom they provided care, were aged 20 to 49 years at the time of data collection. Participants resided in Vienna, Austria and nearby provinces, representing a range of caregiving experiences. Household sizes ranged from two to four members. Nine participants were married, three were divorced, and one was single. Employment status varied: five participants worked part-time, one worked full-time, four were retired, and three were homemakers. Additional details about the individual family situations are provided in Table 2.

### 3.2. Challenges of Aging Parents with Care Responsibilities

The analysis of the data from the group discussion revealed six key themes that capture the experiences and perspectives of ageing parents with care responsibilities: 1. mastering complexity, 2. being an expert and an advocate, 3. balancing autonomy and care, 4. caring as a lifelong journey, 5. standing on the margins of society, and 6. worrying about the future.

#### 3.2.1. Mastering Complexity

This theme highlighted the intricate balancing act that carers have to perform as they navigate the complexities of their responsibilities while striving to provide effective care. Mastering complexity describes the variety of challenges parents faces daily. Planning and structuring each day, providing personal care, and managing various appointments are constant components of their lives. Dealing with assistive devices is a central aspect of life and organization. These devices are perceived as both a support and a necessity for providing appropriate care. Family caregivers need to adapt to the use of technology and devices and integrate them into their daily lives. While Mrs. Pichler and other participants discussed the effort required to transport wheelchairs and accessories (Figure 1), Mrs. Holzer talked about the home ventilator her son depended on (Figure 2). The challenges are highly specific and individual, depending on the requirements and needs of their children.


*“I do his rehabilitation equipment and maintenance and everything. And it has to be done every year. You can see that. So, my car is full to the brim. Every time.”*
(Mrs. Pichler, 51)


*“[…] this is the machine that actually rules our lives.”*
(Mrs. Holzer, 55)

*Mastering complexity* involves not only the individual actions focused on the child but also the challenges of managing the consequences and integrating them into daily life. One significant consequence is the pervasive feeling of isolation and loneliness that many parents experience. It is noteworthy that many parents used the phrase “lockdown as a permanent state,” a term popularized during the pandemic, to describe their ongoing experiences. This sentiment was exemplified by Mr. Zangerl (Figure 3).


*“I’m actually already used to the lockdown. It’s nothing new for me, we’ve had it for 20 years.”*
(Mr. Zangerl, 66)

#### 3.2.2. Being an Expert and Advocate

Parents navigate a complex array of roles, serving as caregivers, partners, and advocates for their children with care needs. They bear the responsibility for all organizational and medical matters, often without professional support. The expectations they hold for themselves, as well as those imposed by others, have driven them to adopt and sustain specific roles, particularly in the realm of care. This often involves a continuous effort to expand their knowledge and skills. Many parents perceive themselves as the sole individuals capable of understanding and anticipating their children’s needs accurately and responding effectively. This sentiment is illustrated in the following quote:


*[…] because [Figure 4] also shows where we feel the strain. One aspect is the responsibility for George’s health. It starts with recognizing when he is unwell. It’s very difficult for him to say where he’s in pain. For example, he often says he has a stomachache, but he really has a headache. It’s very challenging.”*
(Mrs. Brunner, 66)

**Figure 4 ijerph-22-01297-f004:**
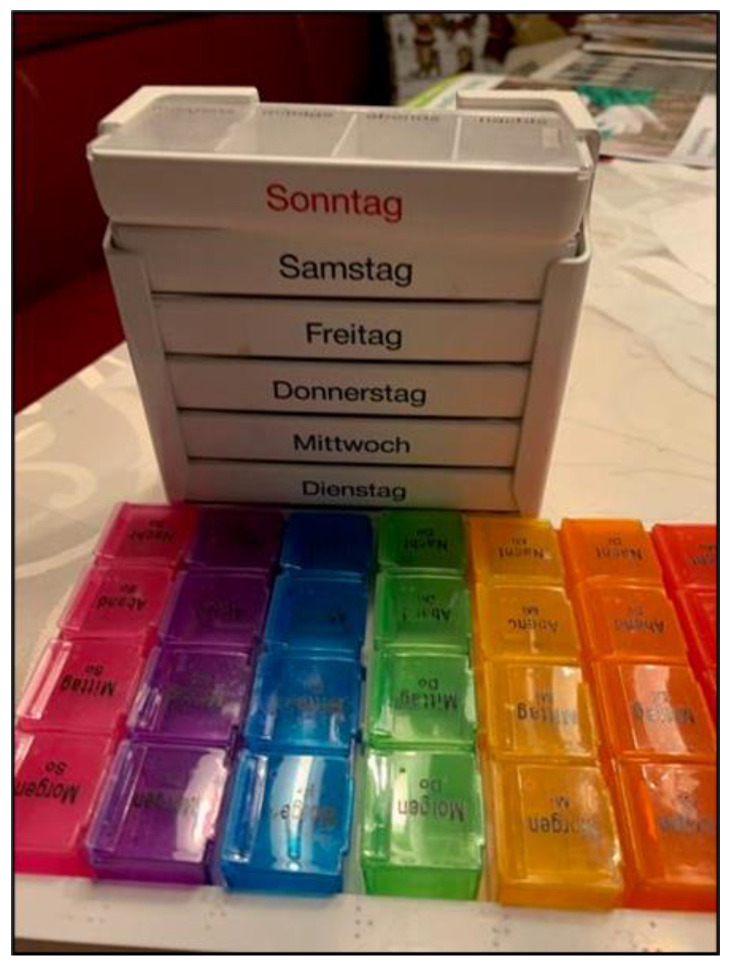
A medicine dispenser as a symbol of various conditions and symptoms and the challenging expert role (Mrs. Brunner, 66).

This dynamic has transformed parents into experts on their children’s needs. To ensure that their children are acknowledged and heard, parents increasingly take on the role of advocates for their children. In this advocacy capacity, they speak out and fight for their children’s rights and needs. Mrs. Pichler and other participants shared several experiences in hospitals and interactions with medical staff that had not been confidence-building and negative outcomes had only been avoided due to their interventions. The following quote from Ms. Pichler shows that, from the parents’ point of view, actions are being imposed on the child without taking the child’s perspective into account (Figure 5).


*“If Sebastian had been alone, without his mother or father, they [staff the hospital] would have tried out things. Sebastian’s quality of life would have been compromised. It would have been over.”*
(Mrs. Pichler, 51)

Expertise also encompasses the challenges parents face in navigating bureaucratic systems and regulations. From their perspective, the bureaucratic demands are substantial. Tasks such as obtaining permits, filling out prescription forms, securing expert reports, and submitting applications are often extremely time-consuming and draining (Figure 6). Parents have to independently navigate what they perceive as a complex, inflexible, and bureaucratic care system to access necessary support services and adequately meet their children’s needs.


*“We have a jungle of prescriptions, health insurance approvals, specialist appointments, special examinations, so we have up to thirty appointments a year. You wouldn’t believe it, but after twenty-two years I still have to fight for this fucking approval every year.”*
(Mrs. Fuchs, 53)

#### 3.2.3. Balancing Autonomy and Care

This theme highlights the multifaceted roles that parents assume, underscoring their dedication to understanding their children’s needs and advocating their well-being within complex systems. Aging parents with caregiving responsibilities often find themselves navigating a delicate balancing act between granting their children freedom and providing necessary care. Parents strive to support their adult children in taking on as much responsibility as possible and making their own decisions, thereby maximizing their freedom. However, the need for care arising from illness or disability brings unique fears and concerns for parents regarding their children’s well-being. The responsibility they feel for their children’s safety often conflicts with their desire to promote autonomy. Despite these fears, parents frequently work to overcome them, striving to provide their children with as much freedom as possible (Figure 7).


*“He’s very proud that he can climb because no one expects him to be able to. But I can’t watch; I only know it from photos.”*
(Mrs. Holzer, 55)

Parents found that, in healthy children, autonomy developed naturally through gradual steps toward independence. In contrast, for children in need of care, the process of relinquishing responsibility had to be carefully assessed, initiated, and guided by the parents themselves. As these children grew, they remained children in the eyes of their parents. Mrs. Riegler eloquently captured this sentiment in her reflections (Figure 8).


*“On the one hand, he’s often like a baby, but on the other hand, he’s often like a grown man. So baby toy, baby, the beer bottle, adult.”*
(Mrs. Riegler, 56)

#### 3.2.4. Care as a Lifelong Journey

This theme underscores the profound impact of caregiving on parents’ lives, highlighting the emotional and physical toll it takes over the years. It also reflects the resilience and dedication that characterizes their lifelong commitment to care. For many parents, the responsibility of caregiving extends throughout their lives, making it a lifelong journey. This role often demands 24/7 commitment, profoundly influencing their daily routines and priorities. As Mrs. Brunner reflected,


*“It starts in the morning with shaving, brushing his teeth, sometimes he resists it, he doesn’t like it that much. So he can’t do it all on his own. Someone always has to be there and as long as our sons and daughters are at home, as long as George is at home, we are responsible for these things all by ourselves.”*
(Mrs. Brunner, 66)

Mrs. Brunner highlighted that she and her husband have been caregivers for 40 years, emphasizing the enduring lack of freedom and flexibility in their everyday live. While friends of the same age had retired and embraced new opportunities, their lives remained unchanged (Figure 9). This stark contrast reinforces how parents with caregiving responsibilities experiences a different reality, particularly as they age.

As the duration of caregiving increases, the limitations on flexibility and recreational opportunities become increasingly burdensome, draining their energy. This constant vigilance means that they can never fully relax or take a break. Parents described a perpetual state of alertness (Figure 10), as illustrated by Mrs. Holzer (55), who stated that


*“Actually, I’m on night duty every day and I’m there immediately if there’s an alarm […]. I have a bed next to his bed, so I almost don’t have to get up […] Being really relaxed doesn’t really exist. There hasn’t really been anything like that for a long time.”*
(Mrs. Holzer, 55)

#### 3.2.5. Standing on the Margins of Society

This theme underscores the isolation and lack of understanding that many caregiving parents face as they navigate a society that often fails to accommodate their unique needs and perspectives. Aging parents with caregiving responsibilities often feel that they and their children occupy the fringes of society. Their children’s differences make them stand out in everyday life, attracting unwanted attention and stares. Parents expressed a sense of not being perceived as equals by society. As Mrs. Pichler (51) poignantly stated,


*“Sebastian is often stared at like a circus animal. I can’t handle that every day either […]. Instead of offering help, they just look on like idiots.”*
(Mrs. Pichler, 51)

While strangers in public often stare shamelessly, other parts of society tend to overlook the challenges faced by parents and their children. Politics, the labor market, architecture, and transportation often fail to adequately consider the situations of caregiving families. Architectural barriers and a lack of awareness regarding spatial obstacles for people with disabilities in public spaces further exacerbate the daily struggles of caregiving parents. Mrs. Schneider recounted an experience she had on public transportation, namely that the elevator to the train platform was not working, which is supposed to be indicated by the small yellow sign in front of the elevator door(Figure 11):


*“A drastic experience for me was when we were traveling on public transport […] and the elevator broke down. This led to my 23-year-old child, who is over one meter eighty […], sitting on the floor and starting to scream and I didn’t really know how to get her home from there.”*
(Mrs. Schneider, 54)

Many caregiving parents encounter negative reactions and even hostility in their daily lives, repeatedly reinforcing their sense of marginalization within society. They often find that their expertise and knowledge regarding their children’s impairments are not acknowledged, and they are shown little empathy or sensitivity in their interactions with others. Mrs. Schneider and Mr. Zangerl expressed their frustration with insensitive comments from authorities. For example, Mrs. Schneider recounted that


*“‘Yes, you brought the child into the world,’ from the youth welfare officer really blew my mind.”*
(Mrs. Schneider, 54)

Similarly, Mr. Zangerl (52) shared that


*“The admin officer told me, ‘They (the parents) want to get rich off their daughter’s disability.’*
(Mr. Zangerl, 52)

#### 3.2.6. Worrying About the Future

This theme illuminates the significant concerns that aging parents with caregiving responsibilities encounter, underscoring the convergence of emotional, physical, and financial challenges as they reflect on their own future and that of their children. Parents with long-term caregiving responsibilities often experience a gradual decline in energy and strength as their caregiving journey continues. This, combined with their own aging, heightens their awareness of personal limits, as illustrated by Mrs. Ebner’s reflection (Figure 12):


*“We were on the Camino de Santiago, walking through Spain and Portugal with a balloon tied to our backpacks. After a week, not only were we exhausted, but so were the balloon. In real life, I find that as you get older, you gradually run out of energy. That’s just the way it is.”*
(Mrs. Ebner, 67)

In addition to physical and emotional exhaustion, many parents face financial pressures due to therapies, therapeutic products, and assistive devices that are not covered by insurance. Caregiving responsibilities often limit their ability to work full-time, with some parents able to work only part-time or not at all. This financial burden is particularly acute for single parents, who found their financial situation to be increasingly precarious. As they looked to the future, many parents expressed deep concern about the possibility of poverty in old age.

The primary concern for these parents is the future of their children and the potential challenges they might face when the parents are no longer able to provide care. A major concern for aging parents is how their children would be supported after their own deaths, as Ms. Strasser (52) expressed (Figure 13):


*“The prospect of my own death is my daily concern. What will happen to my son and what to my daughter who need so much support?”*
(Mrs. Strasser, 52)

Although many parents acknowledged the necessity of establishing arrangements for their children’s future care, the participants did not discuss any concrete plans in this regard. The majority expressed a strong desire to ensure that their children received care that was independent of other family members, yet they frequently reported feelings of uncertainty regarding the means by which this could be achieved. This uncertainty highlights the ongoing challenge of planning for their children’s future well-being.

## 4. Discussion

As far as we know, this is the first study of its kind to use photovoice to draw a picture of the situation of aging parents with caregiving responsibilities. In line with previous research, our findings illustrate how ageing parents of children with care needs must navigate an interwoven set of challenges, in which the complexity of the care situation [8], feelings of invisibility within society [8,25,26], and the delicate balance between supporting autonomy and ensuring adequate care [27,28] are deeply interconnected and mutually reinforcing.

The originality of the photographs stimulates critical thinking, allows for a deeper exploration of the challenges, and provides insight into the lived experiences of the participants. In this way, the families’ stories of daily caregiving responsibilities, concerns for their children’s daily lives, and their desire to do the best for their families are uniquely and movingly portrayed.

Building on this, parents with caregiving responsibilities define their role as giving their children a voice in the world. They fully embrace this meaningful role, shaping their daily lives around it. This bond, however, is inevitably tied to the question of finiteness as both parents and children grow older. As they age, parents find themselves caught between the multifaceted realities of lifelong caregiving and their increasing worries about the future. Both themes, even when considered separately, reveal that the progression of the life course ultimately brings an end to the constant responsibility of caregiving—both for parents and their children.

The theme *Being an Expert and Advocate* highlights one of the multifaceted roles that aging parents assume in caring for their adult children with disabilities. Parents navigate complex systems, often without professional support, and become both experts in their children’s needs and advocates for their rights. This aligns with the findings of Casey et al. [29] on the experiences of Irish mothers, which emphasize the critical role parents play in supporting decision-making for adults with intellectual disabilities. The ongoing efforts of parents to expand their knowledge and skills, as described by our participants, echo Burke et al.’s [30] work on parent advocacy across the lifespan. Parents’ perception of themselves as uniquely capable of understanding and responding to their children’s needs transforms them into experts, a phenomenon also noted by Curryer et al. [27] in their study of mothers supporting the self-determination of adult children with intellectual disabilities.

The advocacy role that parents play in speaking up and fighting for their children’s rights is particularly important in healthcare settings, as illustrated by our participants’ experiences with medical staff. This aligns with the findings of Petner-Arrey and Copeland [31] on the importance of care and advocacy in promoting autonomy for adults with intellectual disabilities. The time-consuming and exhausting nature of administrative tasks adds another layer to their caregiving responsibilities, underscoring the need for more streamlined support systems. Importantly, the concept of parents as advocates is not rigid but fluid, adapting to the changing autonomy of their children. As our study demonstrated, supporting children’s decision-making processes involves more than simply presenting options; it requires ensuring an appropriate understanding of participation in the evolving process of child development. This dynamic approach to advocacy and expertise reflects the ongoing nature of caregiving for aging parents, emphasizing the need for flexible, responsive support systems that recognize and value parental expertise while simultaneously promoting the autonomy of adults with disabilities.

The theme *Worrying About the Future* highlights the deep anxiety that aging parents experience regarding the care of their adult children with disabilities. This concern is well documented in the literature, which identifies uncertainty and a lack of future planning as significant stressors for caregivers [8,9]. This is often associated with the experience that professional support has often been found to be not very helpful [32]. Anderson-Kittow et al. [33] and Forrester-Jones [34] found that many aging parents lack a contingency plan for situations in which they might be unable to provide care, even temporarily, exacerbating their anxiety about the future. This lack of concrete planning is reflected in our study, as participants expressed deep concern about their children’s future care but did not mention specific arrangements. The financial strain experienced by caregivers, particularly lone parents, echoes the findings of Heller et al. [7], who highlighted the negative impact of caregiving on employment and financial stability. The research by Pryce et al. [35] further underscores the financial vulnerability of aging caregivers, noting that the long-term financial impact of caregiving can lead to poverty in old age—a concern explicitly expressed by our participants. The desire to secure care independent of other family members, coupled with uncertainty about how to achieve this, aligns with findings by Taggart et al. [10], who identified the lack of appropriate care options as a significant source of stress for aging caregivers.

Furthermore, there remains a persistent gap between caregivers’ aspirations for their children’s future care and the support structures currently available, underscoring the need for more comprehensive and flexible care options. Kruithof et al. [36] discuss structured group discussions as a tool for planning care for people with profound intellectual and multiple disabilities, emphasizing the importance of such interventions when parents are no longer present. Similarly Brennan et al. [37] explore the health and well-being of sibling carers, revealing the compounded stress and health impacts experienced by those who assume caregiving roles for their siblings with disabilities. These findings collectively emphasize the urgent need for targeted interventions and policy changes to address the complex needs of aging caregivers and their adult children with disabilities.

### Strengths and Weaknesses

This study on aging parents with caregiving responsibilities had both strengths and limitations. A major strength was the use of the photovoice method, which culminated in an evaluation to assess whether the project objectives had been met. Eight participants agreed to follow-up telephone interviews. They reported that this approach empowered them, provided insight into their often overlooked challenges, and fostered a sense of community. Participants expressed gratitude for being involved, feeling heard, and hoping for positive changes. Many were even willing to serve as spokespeople to bring their needs to public attention. However, this study was limited by its inability to access specific groups like families without formal support or those with language barriers. These factors could have impacted the parents’ experience of the situation. While digital media enabled the study to proceed during the post-pandemic period, it was less effective for informal exchanges, as participants preferred face-to-face meetings to share experiences or practical caregiving tips. Despite these obstacles, this study successfully highlighted the experiences of aging parents with caregiving responsibilities and provided a platform for them to voice their situations and needs.

## 5. Conclusions

The issue of aging parents with caregiving responsibilities is a significant public health concern, as it involves a growing population facing distinctive health challenges. This group often experiences heightened stress, physical strain, and neglected self-care, all of which can lead to adverse health outcomes. Addressing their needs is essential for maintaining the overall health and well-being of the broader community. Despite their critical role, this often overlooked group faces unique challenges that profoundly impact their quality of life and well-being. As a vulnerable population in research, aging parents with caregiving responsibilities require more focused studies to fully understand and address their complex needs.

The findings of this study highlight the urgent need for comprehensive support systems, including accessible assisted living facilities and innovative care models. These models should integrate a family-centered approach, with special attention paid to the needs of adult children requiring care. We believe that future research should focus on designing and evaluating such tailored support systems to ensure the well-being of aging caregivers while promoting the autonomy and care of their children.

Additionally, we emphasize the importance of systematically evaluating participatory methodologies like photovoice. While this study has begun to explore the strengths and weaknesses of photovoice through its evaluation, further research is needed to ensure that participation is not merely a “label” but a meaningful and empowering process.

By advancing both the development of support systems and the refinement of participatory research methods, future studies can contribute to more effective and inclusive solutions for this vulnerable population.

## Figures and Tables

**Figure 1 ijerph-22-01297-f001:**
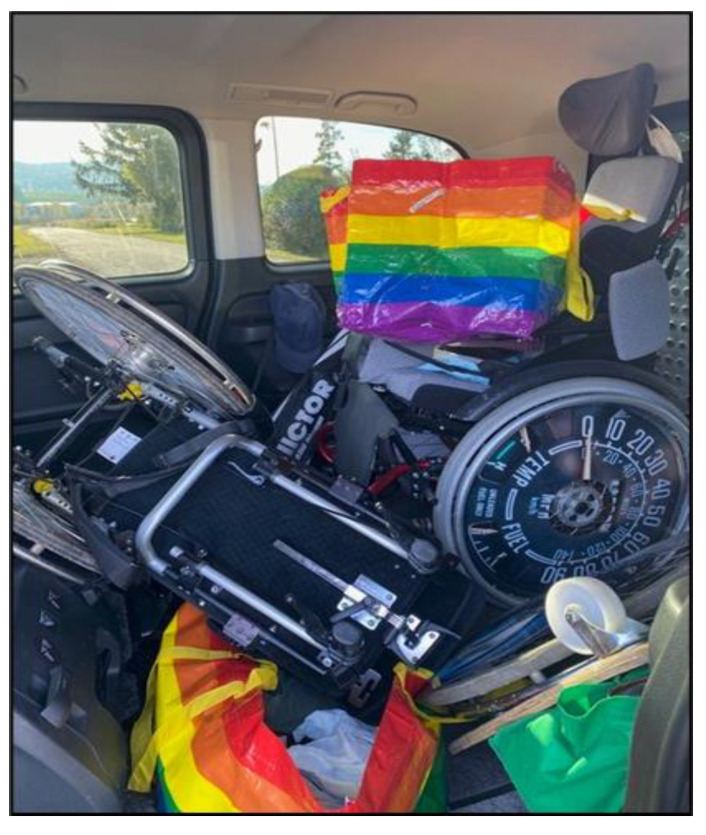
Wheelchairs in a minibus.

**Figure 2 ijerph-22-01297-f002:**
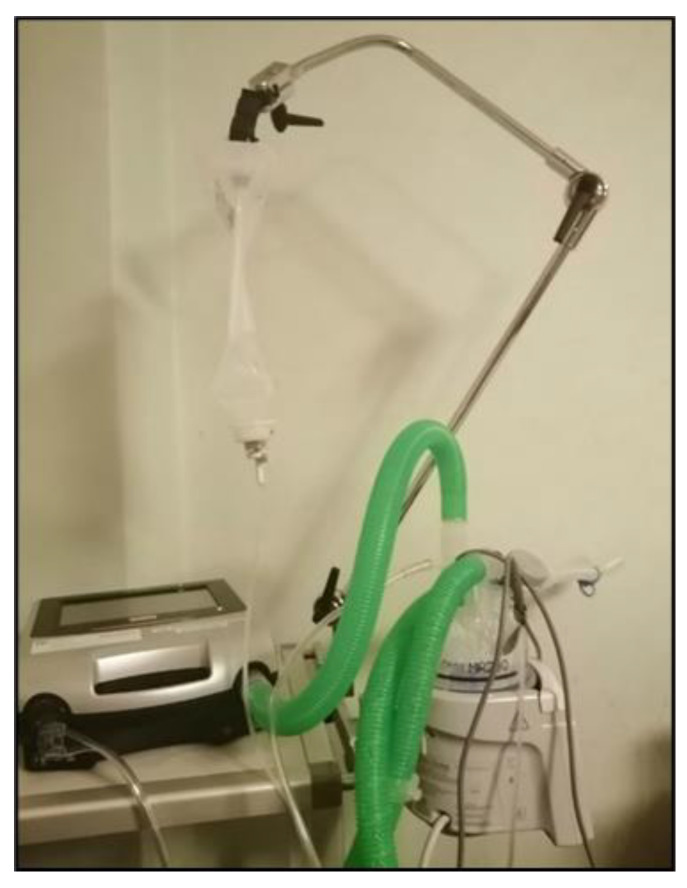
Respirator (Mrs. Holzer, 55).

**Figure 3 ijerph-22-01297-f003:**
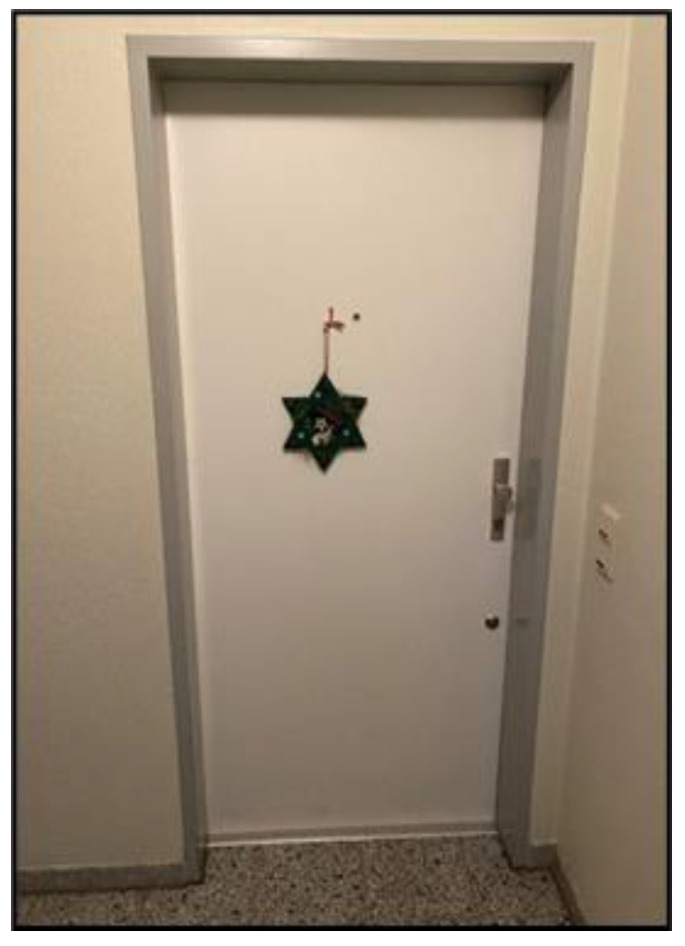
Twenty years of lockdown (Mr. Zangerl, 66).

**Figure 5 ijerph-22-01297-f005:**
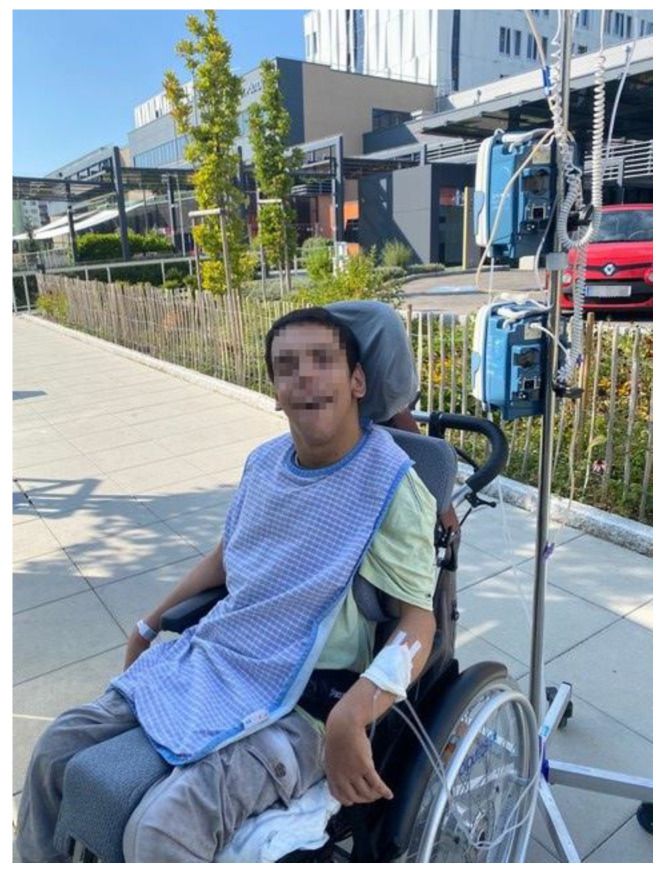
Sebastian in his wheelchair in front of the hospital (Mrs. Pichler, 51).

**Figure 6 ijerph-22-01297-f006:**
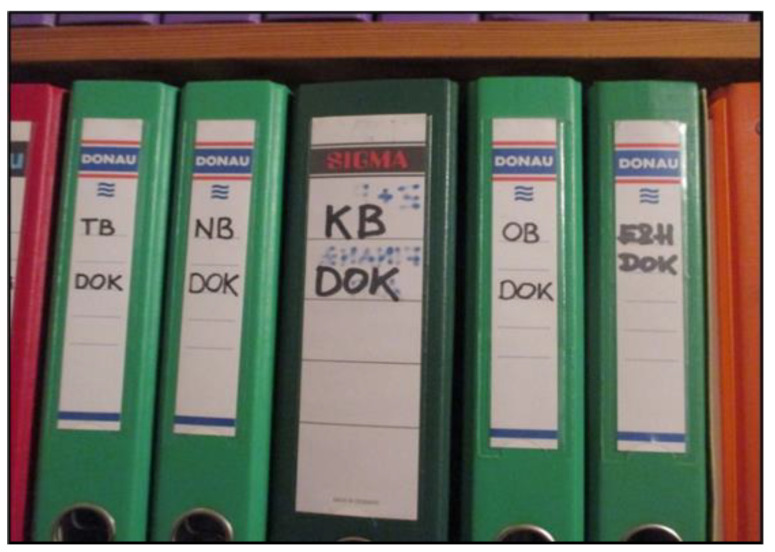
Administrative files (Mr. Lehner, 55).

**Figure 7 ijerph-22-01297-f007:**
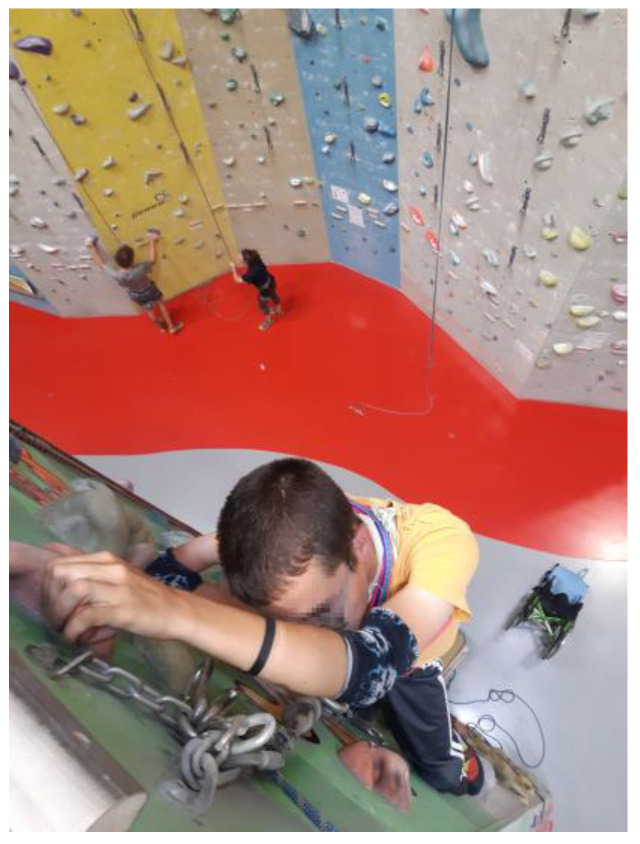
Confidence (Mrs. Holzer, 55).

**Figure 8 ijerph-22-01297-f008:**
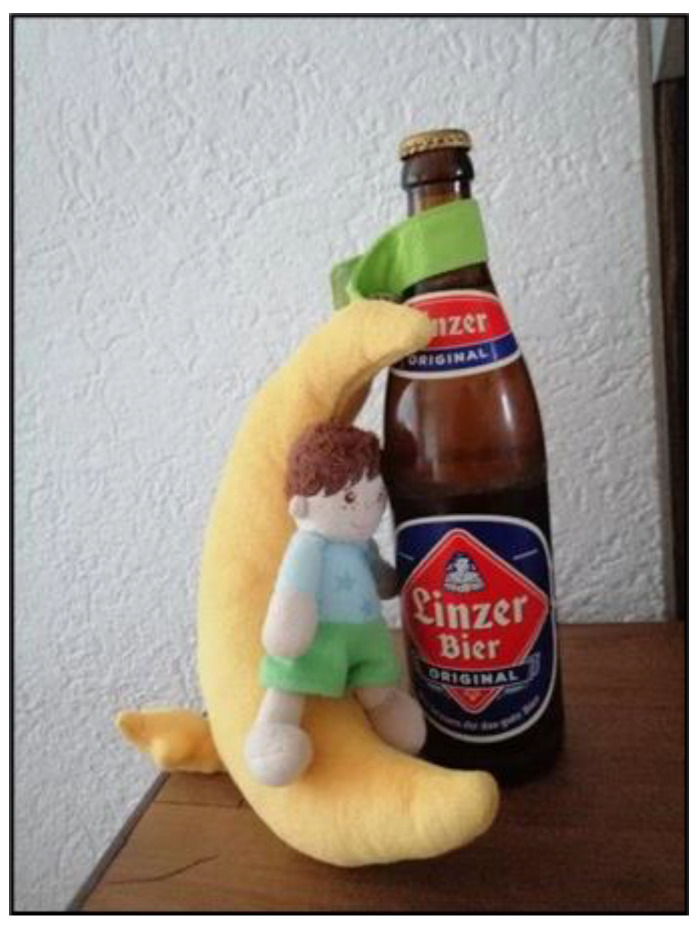
Music box and beer bottle (Mrs. Riegler, 56).

**Figure 9 ijerph-22-01297-f009:**
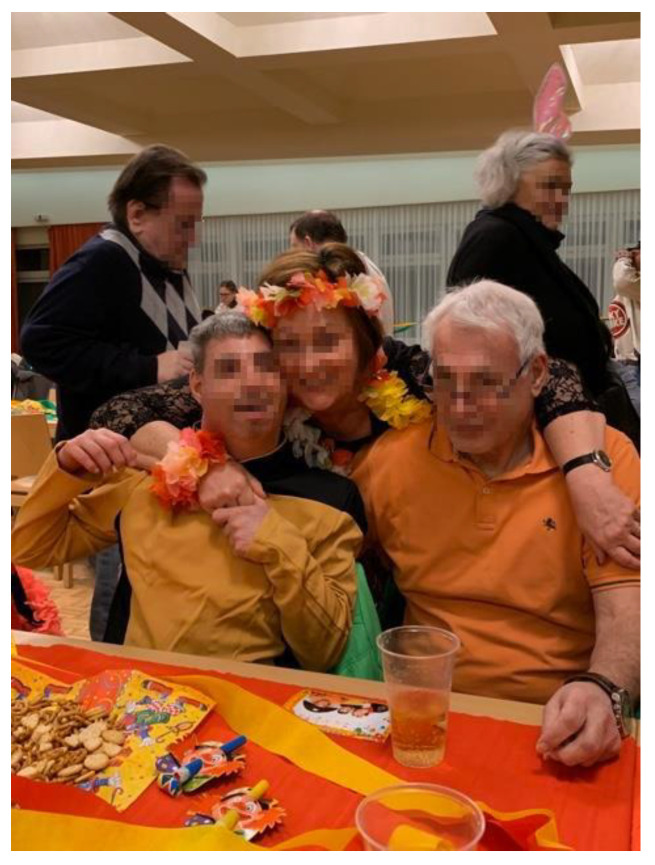
“40 years and we are still here.” (Brunner family).

**Figure 10 ijerph-22-01297-f010:**
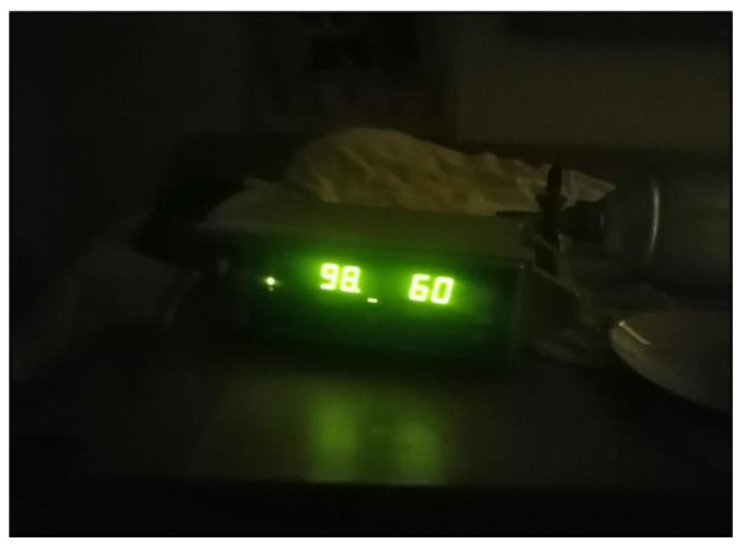
Doing night duty every night (Mrs. Holzer, 55).

**Figure 11 ijerph-22-01297-f011:**
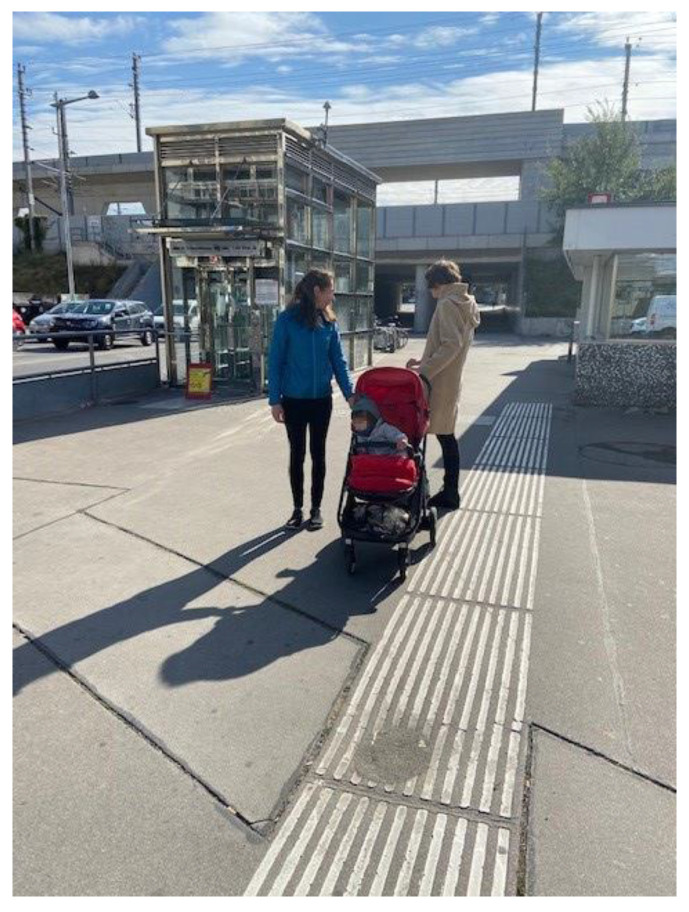
Elevator out of service (Mrs. Schneider, 54).

**Figure 12 ijerph-22-01297-f012:**
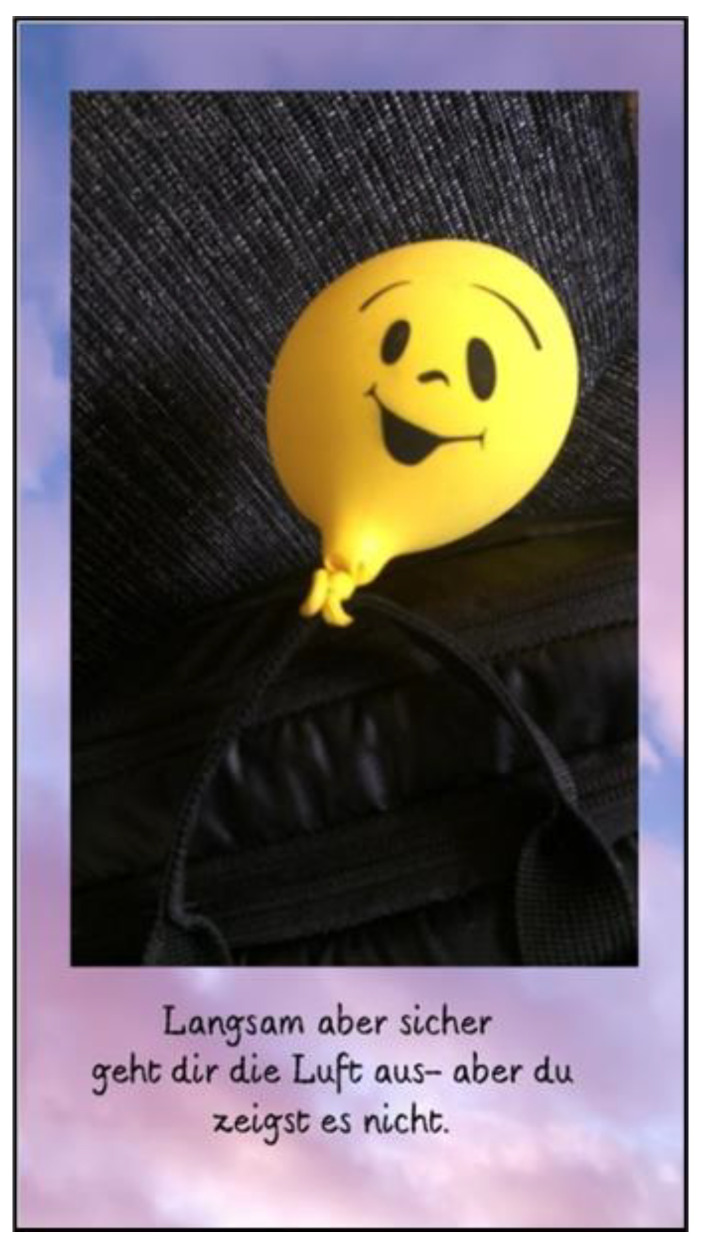
Running out of air (Mrs. Ebner, 67).

**Figure 13 ijerph-22-01297-f013:**
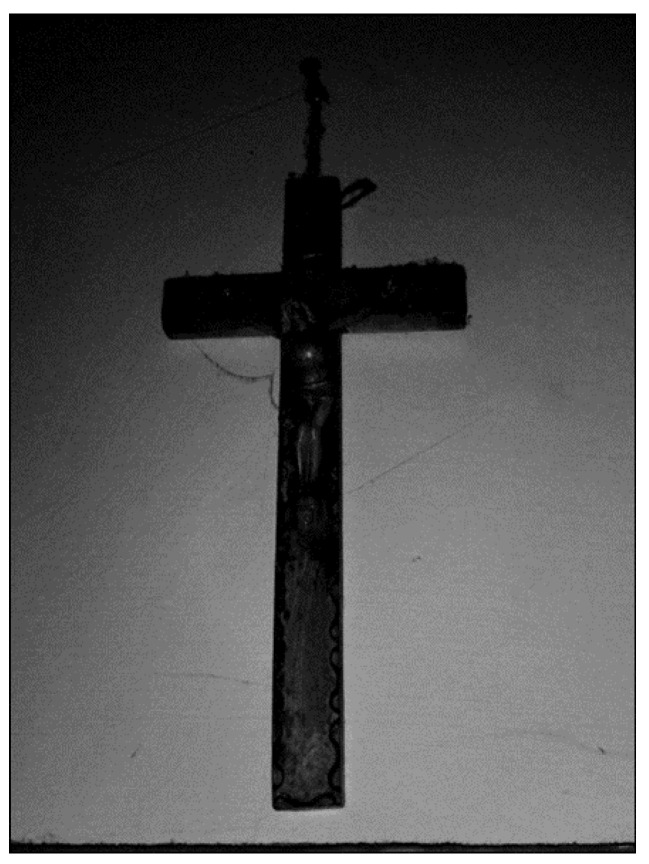
The prospect of her own death (Mrs. Strasser, 52).

**Table 1 ijerph-22-01297-t001:** The showed guide for photovoice technique.

*What do you **S**ee here?* *What is really **H**appening here?* *How does this relate to **O**ur lives?* ** *W* ** *hy does this concern, situation, strength exist?* *How can we become **E**mpowered through our new understanding?* *What can we **D**o about this?*

**Table 2 ijerph-22-01297-t002:** Details of the parents, the children, and the care arrangements. The names have been pseudonymized.

Parents and Their Children	Description of Care Arrangements
1. Mr. Zangerl (52 years) and Anna (21 years)	Mr. Zangerl has been caring for his daughter Anna. Due to physical and mental impairments, Anna requires artificial nutrition. His wife and Anna’s grandmother also contribute to her care, and the family uses a mobile care service for 10 h a month. Anna receives ability-based care Monday through Friday.
2. Mrs. Pichler (51 years) and Sebastian (27 years)	Mrs. Pichler is a single mother who is solely responsible for son Sebastian, who has tetraparesis and communicates using technical aids. During the week, Sebastian resides in a day structure combined with a community home, while Mrs. Pichler manages technology and assistive devices. On weekends, she is completely responsible for his care.
3. Mrs. Riegler (56 years) and Michael (24 years)	Mrs. Riegler cares for her son Michael, who has cognitive impairments from birth complications and requires assistance with all daily activities. Although her mother and husband provide support, her husband has physical limitations that limit his involvement. Michael attends a daycare center during the week and goes out weekly with a personal care attendant.
4. Mrs. Schneider (54 years) and Simona (24 years)	Mrs. Schneider works part-time while also caring for her daughter Simona, who has multiple disabilities caused by a genetic condition. Simona’s challenges include spasticity, autism with Tourette’s syndrome, and nonverbal communication. Mrs. Schneider’s mother assists with Simona’s care from Friday to Saturday. The family also receives medical care services in the early morning and evening. From Monday to Friday, Simona participates in a structured day program.
5. Mrs. Fuchs (53 years) with Johanne (23 years)	Mrs. Fuchs has been the primary caregiver for her daughter Johanne, who lives with a rare chronic illness that requires constant monitoring. Over the years, Mrs. Fuchs has become highly knowledgeable about Johanne’s condition. While a mobile pediatric nurse provided support when Johanne was younger, Mrs. Fuchs now handles all caregiving responsibilities on her own.
6. Mr. Strasser (52 years) and Fabian (27 years)	Mrs. Strasser is the sole caregiver for her son Fabian, who has autism spectrum disorder and developmental disabilities. She receives some support from a day center for his care. In addition, she occasionally assists her adult daughter, who is partially dependent on her help.
7. Mrs. Holzer (55 years) and Matthias (24 years)	Mrs. Holzer has dedicated over 20 years to caring for her son Matthias, who has required nightly ventilation since undergoing brain stem surgery as an infant. Matthias, who uses a wheelchair due to tetraparesis, requires extensive support in his daily life. While Mrs. Holzer receives occasional help from family and friends, she often pays out of pocket for assistants to ensure she can continue working.
8. Mrs. Winter (64 years) and Elisabeth (35 years)	Mrs. Winter is retired and provides full-time care for her daughter Elisabeth, who has a brain malformation, epilepsy, and cognitive impairments, requiring constant support. She receives help from Elisabeth’s father, sister, aunt, neighbors, and an assistance dog. In addition, Elisabeth participates in a structured day program on weekday mornings from Monday to Friday.
9. Mrs. Brunner (66 years) with Georg (42 years)	Mrs. Brunner and her husband share the responsibility of caring for their son Georg, who attends a daycare center on weekdays. The family occasionally makes use of respite services. Georg requires extensive support due to a malformation syndrome, including assistance with personal hygiene, eating, and leisure activities. He has epilepsy and communicates nonverbally.
10. Mrs. Aigner (76 years) with Martina (49 years)	Mrs. Aigner is retired and coordinates the care of her daughter Martina, with help from her granddaughters. Martina benefits from weekly visits with a friend for crafts, as well as physiotherapy, medical care, and psychological support. She has experienced multiple cognitive, psychological, and physical illnesses, some of which have been present since birth.
11. Mr. Lehner (55 years) with Benjamin (20 years)	Mr. Lehner is self-employed and cares for his son Benjamin, with support from his wife, who works full-time. Benjamin has cognitive and psychomotor impairments, epilepsy, and a psychotic disorder. After an extensive search, the family found a suitable daycare program that Benjamin attends on weekdays.
12. Mrs. Ebner (67 years) with Florian (40 years)	Mrs. Ebner is currently dealing with her own illness, which has led to her son Florian temporarily moving into sheltered housing. Florian, who has cognitive and physical impairments due to a congenital disability, requires 24 h care. His father, sister, and niece also contribute to his care.
13. Mrs. Mayer (53 years) with Manuel (26 years)	Mrs. Mayer is the primary caregiver for her son Manuel, who has spastic quadriplegia and significant physical limitations. She organizes weekly leisure activities and exercise therapies to improve Manuel’s quality of life. Balancing her caregiving responsibilities with part-time work, Mrs. Mayer also relies on her partner for additional support in caring for Manuel.

## Data Availability

The anonymized data presented in this study are available on request from the corresponding author. The data are not publicly available due to ethical considerations related the anonymity of the study participants.

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
