# Peer review of "“Slowly but Steadily, You’re Running Out of Steam”: Aging Parents’ Caregiving Experiences Through Photovoice"

_ijerph, 2025, doi:10.3390/ijerph22081297_

Round 1

Reviewer 1 Report

Comments and Suggestions for Authors

The use of Photovoice in this study provides a more visual understanding of the challenges facing parents of disabled individuals which have been previously identified in other studies and is to be commended. It is a well written paper addressing an important issue in the field of intellectual disabilities, however I have some concerns around the confidentiality of some of the participants/disabled individuals which I have addressed in specific questions below.

Materials and Methods:

(line 122) Why was it necessary to obtain consent to use images of identifiable faces in publications? Does this not negate the confidentiality aspect which was being addressed by using pseudonymisation?

Results (line 139) The introduction to this section reads like template advice of what to include. Was this deliberate?

(line 201) I understand the relevance of the quote about the challenges of communication with these individuals. However, I am not sure that Photo 4 corresponds with the quote. It shows an array of medicine boxes illustrating the amount of medical knowledge the carers need, but this is not what the quote is about. I wonder if you could choose another photo which might provide a better illustration of the quote?

(line 211) As per my previous query about having recognisable faces in the photos, I think Photo 5 could have the face pixelated for confidentiality.

(line 214) Is there any way of providing a little more clarity about what this mother meant in this quote? Hospital staff 'trying out things' might not necessarily be a negative thing unless the reader can understand what the mother's concern was.

(line 226) I'm just wondering about the lettering on the files on Photo 6. In the interests of confidentiality, can you confirm that this lettering does not relate tot he individual's real name? If it does, is there any way of pixelating this?

(line 268) Can the faces in Photo 9 be pixelated for confidentiality? If not, the pseudonymisation throughout the paper is redundant.

(line 296) I'm not sure that photo 11 illustrates what the quote is referring to. If this is the only one you have, I would suggest providing a little more explanation in the legend to help the reader understand what they are looking at and its relevance to the quote, as it doesn't appear to be showing a broken elevator.

References: 

Many of the references are quite old. I appreciate that the Wang paper (1997) is a seminal paper regarding Photovoice but perhaps some of the others could be reviewed. You may wish to consider:

Masterson et al (2018) Photovoice for mobilizing insights on human well-being in complex social-ecological systems: Case studies from Kenya and South Africa. Ecology and Society 23/3:13. https://doi.org/10.5751/ES-10259-230313 

Watchman et al (2020) Revisiting photovoice: Perceptions of dementia among researchers with intellectual disability. Qualitative Health Research, 30(7), 1019-1032. https://doi.org/10.1177/1049732319901127 

There are also more recent papers on residential care and ageing carers which may be appropriate.

No 8 Black et al 2024. This link takes the reader to the Positive Futures page but there is no evidence of the report you have referenced. Can you review this please?

No. 12 von Unger 2014. this is an older paper so perhaps find a more recent one. If you choose to use it, please provide an English translation.

Author Response

Comments and Suggestions for Authors

The use of Photovoice in this study provides a more visual understanding of the challenges facing parents of disabled individuals which have been previously identified in other studies and is to be commended. It is a well written paper addressing an important issue in the field of intellectual disabilities, however I have some concerns around the confidentiality of some of the participants/disabled individuals which I have addressed in specific questions below.

Dear Reviewer,

Thank you for your valuable feedback and your transparent comments on data protection. Protecting participants is of paramount importance to us. All participants were fully informed about the significance of the photos and the planned publication, and they provided their consent. In light of the confidentiality concerns raised by all reviewers, we have decided to further enhance data protection by pixelating all photos, even though this deviates from the original agreement with participants.

Materials and Methods:

  1. (line 122) Why was it necessary to obtain consent to use images of identifiable faces in publications? Does this not negate the confidentiality aspect which was being addressed by using pseudonymisation?

Thank you for your critical feedback. In addition to obtaining consent for the use of the photographs, we also applied pseudonymization. This protects additional personal data and reduces the risk of identifying individuals, without significantly compromising the transparency and authenticity of the image documentation. We have carefully balanced these considerations. Nevertheless, due to legitimate concerns regarding data protection, we have decided to pixelate the images (line X):

For reporting purposes, the names and personal data of the participants were pseudonymized to provide protection. For ethical reasons, all faces in the photographs were pixelated to protect the participants' identities in the publication.

  1. Results (line 139) The introduction to this section reads like template advice of what to include. Was this deliberate?

Thank you. We left that by mistake. We have deleted it.

  1. (line 201) I understand the relevance of the quote about the challenges of communication with these individuals. However, I am not sure that Photo 4 corresponds with the quote. It shows an array of medicine boxes illustrating the amount of medical knowledge the carers need, but this is not what the quote is about. I wonder if you could choose another photo which might provide a better illustration of the quote?

Thank you very much for your feedback. We agree that this is not clear. The medicine box symbolizes a pragmatic approach to treatment, which is not reflected in parents' everyday lives. This contrast highlights the photos' multidimensional meaning and narrative function. We have decided to keep this photo but we gave more context information and clearer explanation “Photo 4: Medicine dispenser as a symbol of various conditions and symptoms and the challenging expert role” (L232)

  1. (line 211) As per my previous query about having recognisable faces in the photos, I think Photo 5 could have the face pixelated for confidentiality.

We understand your concerns, and as previously mentioned, we will pixelate all images.

  1. (line 214) Is there any way of providing a little more clarity about what this mother meant in this quote? Hospital staff 'trying out things' might not necessarily be a negative thing unless the reader can understand what the mother's concern was.

Thank you for sharing your critical perspective. We agree that this is unclear. This paragraph clearly illustrates the dynamics experienced by parents. While not necessarily negative, it refers to actions imposed on the child in certain situations from the parents’ perspective without adequately considering the child’s viewpoint. We have added an additional sentence before the quote that hopefully clarifies what was meant. “The following quote from Mrs. Pichler shows that, from the parents' point of view, actions are being imposed on the child without taking the child's perspective into account.” (L243)

  1. (line 226) I'm just wondering about the lettering on the files on Photo 6. In the interests of confidentiality, can you confirm that this lettering does not relate tot he individual's real name? If it does, is there any way of pixelating this?

All names in this study have been pseudonymized. The authors know the real names. Since the use of photos with faces obviously causes great discomfort, we have decided to pixelate the faces. 7. (line 268)

  1. Can the faces in Photo 9 be pixelated for confidentiality? If not, the pseudonymisation throughout the paper is redundant.

We understand your concerns, and as previously mentioned and have pixelated all images with faces.

  1. (line 296) I'm not sure that photo 11 illustrates what the quote is referring to. If this is the only one you have, I would suggest providing a little more explanation in the legend to help the reader understand what they are looking at and its relevance to the quote, as it doesn't appear to be showing a broken elevator.

Thank you very much! The sign in front of the elevator is difficult to see. It indicates that the elevator is not wheelchair accessible (line X: Elevator out of service).  We have tried to make this aspect clearer with the following sentence before the quote: “Mrs. Schneider recounted an experience she had on public transportation, namely that the elevator to the train platform was not working (which is supposed to be indicated by the small yellow sign in front of the elevator door)” (L329)

References: 

  1. Many of the references are quite old. I appreciate that the Wang paper (1997) is a seminal paper regarding Photovoice but perhaps some of the others could be reviewed. You may wish to consider:

Thank you for taking the time to provide us with additional sources in which the Photovoice method has been applied. After careful consideration we have decided to include an alternative source (systematic review from 2023) that has a special focus on people with intellectual disabilities (Ref nr 16, L 73)

Chinn, D., & Balota, B. (2023). A systematic review of photovoice research methods with people with intellectual disabilities. Journal of Applied Research in Intellectual Disabilities, 36(4), 725–738. https://doi.org/10.1111/jar.13106

  1. There are also more recent papers on residential care and ageing carers which may be appropriate.

Thank you for your helpful suggestion. We have included three additional sources.

Introduction (L54)

 Geuze, L.; Goossensen, A.; Schrevel, S. “Continuously struggling for balance”: The lived experiences of Dutch parents caring for children with profound intellectual and multiple disabilities. J. Intellect. Dev. Disabil. 2022, 1-11 10.3109/13668250.2022.2073707

Discussion (L431)

“This is often associated with the experience that professional support has often been found to be not very helpful“: Jordan, J., Larkin, M., Tilley, E., Vseteckova, J., Ryan, S., & Wallace, L. (2025). TransitionsRelated Support for Ageing Family Carers of Older People With Intellectual Disabilities Who Convey Behaviours That Challenge Others: A Systematic Rapid Scoping Review. Journal of Applied Research in Intellectual Disabilities, 38(1), e13322-n/a https://doi.org/10.1111/jar.13322

Discussion (line 386)

“and the delicate balance between supporting autonomy and ensuring adequate care [27, 28] are deeply interconnected and mutually reinforcing. Walker, R.; Hutchinson, C. Care-giving dynamics and futures planning among ageing parents of adult offspring with intellectual disability. Ageing Soc. 2019, 39, 1527 10.1017/S0144686X18000144.

  1. No 8 Black et al 2024. This link takes the reader to the Positive Futures page but there is no evidence of the report you have referenced. Can you review this please?

Thank you for bringing this to our attention. We have removed the source because it is no longer available. We now refer to the following article regarding the uncertainty about future of the children. (Source Nr 8)

Fernández-Ávalos, M. I., Pérez-Marfil, M. N., Ferrer-Cascales, R., Cruz-Quintana, F., Clement-Carbonell, V., & Fernández-Alcántara, M. (2020). Quality of Life and Concerns in Parent Caregivers of Adult Children Diagnosed with Intellectual Disability: A Qualitative Study. International Journal of Environmental Research and Public Health, 17(22), 8690. https://doi.org/10.3390/ijerph17228690

  1. No. 12 von Unger 2014. this is an older paper so perhaps find a more recent one. If you choose to use it, please provide an English translation.

Thank you very much for your feedback. Because of its detailed and well-elaborated description of the Photovoice method, we consider this work a fundamental reference for our study and would like to keep it in our bibliography.

von Unger, H., Partizipative Forschung. Einführung in die Forschungspraxis. [Participatory Research: Introduction to Research Practice]. Springer: Wiesbaden, 2014.

Reviewer 2 Report

Comments and Suggestions for Authors

The study has the potential to be published; its subject is original, its methodology is current, and its message is impactful. However, there are significant methodological and structural deficiencies, particularly in the introduction, methods, and discussion sections. Furthermore, the proposed contributions of the study are not sufficiently clearly presented, and the structure between themes remains weak. For all these reasons, the article will be publishable only if significantly improved.

1. Although the title begins with a compelling metaphor, it is excessively long and lacks academic clarity. A shorter, more focused title is recommended.

2. The abstract is structurally weak. The purpose, method, findings, and conclusions should be clearly distinguished; the main findings should be written more emphatically.

3. The literature is not sufficiently up-to-date in the introduction. More emphasis should be placed on meta-analyses or systematic reviews conducted after 2020.

4. A conceptual framework for the Photovoice method is not presented in the introduction, and the rationale for choosing this method should be theoretically based.

5. The research question is clearly stated, but the unique contribution of the study is not emphasized.

6. The Method section does not specify why the sample size (n=13) was deemed sufficient and how data saturation was achieved.
7. Although participant demographics are detailed, variables such as ethnicity, income level, and immigration status are missing, limiting generalizability.
8. While the thematic analysis process is explained step-by-step, code examples and how themes were generated are not presented in sufficient detail.
9. Images were used as an effective tool; however, photo descriptions are inadequate in some places, and ethical regulations should be presented in more detail.
10. The discussion section focuses on only two themes, while other themes are not included in the discussion. This weakens the integrity of the analysis and discussion.
11. The connection to the literature is often superficial, and the emphasis on original contributions is lacking. An analytical synthesis of the findings should be developed; not simply stating "parallels."
12. The conclusion section is a repetition of the discussion section. It lacks a unique closing structure, and the recommendations remain abstract.
13. The study does not include the perspectives of individuals with disabilities themselves; this deficiency should be recognized and suggestions for future research should be presented in the conclusion section.

Author Response

Comments and Suggestions for Authors

The study has the potential to be published; its subject is original, its methodology is current, and its message is impactful. However, there are significant methodological and structural deficiencies, particularly in the introduction, methods, and discussion sections. Furthermore, the proposed contributions of the study are not sufficiently clearly presented, and the structure between themes remains weak. For all these reasons, the article will be publishable only if significantly improved.

Dear Reviewer,

Thank you very much for your thoughtful feedback. We appreciate your recognition of the study's originality and potential significance. We have addressed your concerns to the best of our ability.

  1. Although the title begins with a compelling metaphor, it is excessively long and lacks academic clarity. A shorter, more focused title is recommended.

Thank you for this feedback. We have tried to find a shorter and more academic title: “Fading Strength: Aging Parents' Caregiving Experiences Through Photovoice”. However, during the communicative validation with the participants (please see section 5. Trustworthiness), they emphasized the importance of highlighting this aspect. Therefore, we aim to retain the metaphor while striving for a shorter and clearer statement regarding the method. Here is the suggestion of the title: "Slowly but Steadily, You're Running Out of Steam”. Aging Parents' Caregiving Experiences Through Photovoice

  1. The abstract is structurally weak. The purpose, method, findings, and conclusions should be clearly distinguished; the main findings should be written more emphatically.

Thank you for this comment. We have reorganized the structure of the abstract to meet your comments. With the introduction and conclusion, we have tried to incorporate the topic into a public health debate.

  1. The literature is not sufficiently up-to-date in the introduction. More emphasis should be placed on meta-analyses or systematic reviews conducted after 2020.

Thank you for this very important comment. We went into to literature (including review articles) again and tried to update the literature based on our arguments in the introduction.

- Burden on caregivers' physical, mental, and financial resources (L35)

Dückert, et al. Multidimensional Burden on Family Caregivers of Adults with Autism Spectrum Disorder: a Scoping Review. Multidimensional Burden on Family Caregivers of Adults with Autism Spectrum Disorder: a Scoping Review 2023, 10.1007/s40489-023-00414-1.

- Vulnerability due to social and economic deprivation, negatively impact on financial status, risk of physical and mental health problems (L53)

Santos, T. et al: Caregiver support, burden, and long-term planning among caregivers of individuals with intellectual and developmental disabilities: A cross-sectional study. Caregiver support, burden, and long-term planning among caregivers of individuals with intellectual and developmental disabilities: A cross-sectional study 2023, 36, (6), 1229-1240 https://doi.org/10.1111/jar.13141

- Lack of suitable support structures (L57)

Kanthasamy, et al.  Family Caregiver Adaptation during the Transition to Adulthood of Individuals with Intellectual Disabilities: A Scoping Review. Family Caregiver Adaptation during the Transition to Adulthood of Individuals with Intellectual Disabilities: A Scoping Review 2024, 12, (1), https://doi.org/10.3390/healthcare12010116  

- Emotional und physical well-being (L53) and uncertainty about the future (L56)

Fernández-Ávalos, M. I et al.  Clement-Carbonell, V.; Fernández-Alcántara, M. Quality of Life and Concerns in Parent Caregivers of Adult Children Diagnosed with Intellectual Disability: A Qualitative Study. Int. J. Environ. Res. Public Health 2020, 17, (22), https://doi.org/10.1108/QAOA-11-2018-0057.

  1. A conceptual framework for the Photovoice method is not presented in the introduction, and the rationale for choosing this method should be theoretically based.

Thanks again for this comment. We included a theoretical rational of photovoice in the instruction of the method section (L70) and an argument why photovoice was chosen for this study (L78).

  1. The research question is clearly stated, but the unique contribution of the study is not emphasized.

Thank you. We are not sure what is meant with the unique contribution of the study. In the paragraph of the research question, we have tried to highlight the unique contribution in this research with the statement “… insight into this often-overlooked group, empower participants through active engagement in the research process, and foster collective exchange among caregivers.“ (L61)

  1. The Method section does not specify why the sample size (n=13) was deemed sufficient and how data saturation was achieved.

Thank you again for this very important point. Concerning the sample size, it was so that 20 parents showed great interest in participating in the study. Due to a lack of time or illness (n=7) 13 parents finally participated in the study. We have added this information in Planning, Preparation, and Recruitment (L80).

The question of data saturation is a potential weakness of the study. We gained comprehensive insight into the lifeworld of the participants (158 photos, L100). However, we were unable to access parents without formal support. Therefore, we cannot make any statements about this group. However, we believe that we could have covered some topics in greater depth. We have included this aspect in Strengths and Weeknesses, (L465).

  1. Although participant demographics are detailed, variables such as ethnicity, income level, and immigration status are missing, limiting generalizability.

The question of data saturation is a potential limitation of the study. While we gained comprehensive insight into the lifeworld of the participants (158 photos, L100), we were unable to include parents without formal support. As a result, we cannot make any statements about this group. Additionally, we believe that some topics could have been explored in greater depth (see Section Strengths and Limitations, L427)

  1. While the thematic analysis process is explained step-by-step, code examples and how themes were generated are not presented in sufficient detail.

Thank for this comment. Please see Appendix A for a code example.

  1. Images were used as an effective tool; however, photo descriptions are inadequate in some places, and ethical regulations should be presented in more detail.

We have attached particular importance to these aspects. With regard to the inadequate description in some places, we have revised this on the basis of specific examples from the other reviews. Please see lines 236, 243, 329

Regarding ethics, we are unsure which aspects may have been insufficiently described. After reviewing other studies, we believe we have adequately addressed the ethical considerations. That said, we agree that a photo study raises many ethical challenges. We presented these in detail to the ethics committee and decided to pixelate all photos in which study participants can be identified.

  1. The discussion section focuses on only two themes, while other themes are not included in the discussion. This weakens the integrity of the analysis and discussion.

Thank you for raising this point, as we thought long and hard about how to structure the discussion beforehand. We have now included the other aspects of the study at the beginning of the discussion, taking into account the existing literature e.g. Curryer et al 2020 and Walker et al. 2019 (L387)

We decided to maintain an in-depth discussion of these two aspects because we believe they are central to the study. In the third paragraph (Building on this, parents with caregiving responsibilities define their role as giving their children a voice in the world (...)), we feel we have already introduced these points. (L394-401)

  1. The connection to the literature is often superficial, and the emphasis on original contributions is lacking. An analytical synthesis of the findings should be developed; not simply stating "parallels."

Thank you for this point. We believe that the paragraph "Building on this, parents with caregiving responsibilities define their role as giving their children a voice in the world. (...)" has the potential to do justice to this synthesis because parents see caregiving as a vital, lifelong role that gives their children a voice, but as both age, this commitment becomes intertwined with growing concerns about the future and the inevitability of caregiving eventually coming to an end. We hope you agree.

  1. The conclusion section is a repetition of the discussion section. It lacks a unique closing structure, and the recommendations remain abstract.

Thank you very much for bringing this point to our attention. Keeping this in mind, we have reorganized the entire conclusion (paragraph 2, 3, and 4, L484-496). We hope it addresses your concerns regarding abstract recommendations and structure.

  1. The study does not include the perspectives of individuals with disabilities themselves; this deficiency should be recognized and suggestions for future research should be presented in the conclusion section.

Thank you for your suggestions. In the conclusion, we tried to address this point by emphasizing the importance of children's perspective on support system arrangements (L484) and we have added another source in the method section as an example of including the perspective of people with intellectual disabilities: Chinn, D., & Balota, B. (2023). A systematic review of photovoice research methods with people with intellectual disabilities. Journal of Applied Research in Intellectual Disabilities, 36(4), 725–738. https://doi.org/10.1111/jar.13106 (L72) In general, the study addressed parents, who were pleased to be recognized as a specific group.

Thank you also for the point “future research”. We have included this in the new Conclusion in Line 487-493

Reviewer 3 Report

Comments and Suggestions for Authors

Given its originality, robust theoretical grounding, methodological rigor, and the relevance of its findings, which contribute meaningfully to both academic knowledge and practical application, my overall appreciation of the manuscript “Slowly but Steadily, You're Running Out of Steam Experiences of Aging Parents with Caregiving Responsibilities Through Photovoice” is very positive.

The study addresses a highly pertinent topic within the field of ageing and disability, using a creative and innovative methodological approach that, as evidenced by the results, enabled a successful and insightful exploration of the topic.

However, for the article to be suitable for publication, I believe there are certain details that require revision and further development, which I outline below.

Regarding formatting and references rules:

- some abbreviated journal names should be revised as they appear with different forms (e.g.: TIZARD LEARN DISABI).

- after the name of the last author there is a comma, which is wrong since it should appear a period.

- the volumes of journals are not italicized.

The Stimulating questions for the photo phase and the Showed guide for photovoice technique are presented as Tables but this information is not in a Table format.

In the figure captions the word appears both as “photo” or “foto”.

Material and methods section is presented with the topics common to projects of this nature, but the following points are lacking further development and clarification:

- in the recruitment process, how were these 13 participants reached? Were there any withdrawals or refusals? what was the inclusion criteria?

- Authors said that “Data were collected between January 2022 and November 2022. During the subsequent two-week photo phase, parents captured 10–15 photographs of their caregiving responsibilities using smartphones or disposable cameras”. It's not clear how it can be 11 months of data collection if it took two weeks the photograph process.

- Please clarify the concept of “video group discussions”. Were they synchronous or asynchronous moments? Each participant presented all their 10-15 photos on that unique session? How was this sharing process?

- To better understand the data dimension, it would be important to have information on the session´s recording time and on the size of the transcripts.

- Given the importance of this step to the process of conducting photovoice, it would be important to have more information regarding the additional meeting conducted with participants to review and confirm the results.

On the limitations of the study, the authors reflect on important features of the application of this methodology, but it could be enriched through the following aspects:

- What is the importance of the final exhibition according to Wang and Burris rationale.

- What the impact of a solo online format in the group dynamic. The group modality is so little considered that if this study had been in an individual format (interviews) it seems that it would be the same thing.

And for both questions, what future studies should consider and improve.  

Once again, I congratulate the authors for such an important study.

Author Response

Comments and Suggestions for Authors

Given its originality, robust theoretical grounding, methodological rigor, and the relevance of its findings, which contribute meaningfully to both academic knowledge and practical application, my overall appreciation of the manuscript “Slowly but Steadily, You're Running Out of Steam Experiences of Aging Parents with Caregiving Responsibilities Through Photovoice” is very positive.

The study addresses a highly pertinent topic within the field of ageing and disability, using a creative and innovative methodological approach that, as evidenced by the results, enabled a successful and insightful exploration of the topic.

However, for the article to be suitable for publication, I believe there are certain details that require revision and further development, which I outline below.

Dear Reviewer,

Thank you so much for your kind words and your constructive feedback on our paper. We have tried to incorporate all your important suggestions into the revised version.

Regarding formatting and references rules:

  1. some abbreviated journal names should be revised as they appear with different forms (e.g.: TIZARD LEARN DISABI).

Thank you very much! We hope that we have now updated the Reference section according to the Journal guideline.

  1. after the name of the last author there is a comma, which is wrong since it should appear a period –

adapted

  1. the volumes of journals are not italicized

adapted

  1. The Stimulating questions for the photo phase and the Showed guide for photovoice technique are presented as Tables but this information is not in a Table format.

Thank you. We have integrated it into the text.

  1. In the figure captions the word appears both as “photo” or “foto”.

Thank you again. We have changed to “photo” at all places (2 photos, no track change)

Material and methods section is presented with the topics common to projects of this nature, but the following points are lacking further development and clarification:

  1. in the recruitment process, how were these 13 participants reached? Were there any withdrawals or refusals? what was the inclusion criteria?

Thank you for this important point. We have integrated this information in Planning, Preparation, and Recruitment:

Included were parents of adult children in need of care who wanted to talk about their situation and were willing to take photos of their situation and share them in a group discussion (L80).

Twenty parents expressed interest in participating in the study. However, due to time constraints or illness, only 13 parents ultimately participated (L86).

  1. Authors said that “Data were collected between January 2022 and November 2022. During the subsequent two-week photo phase, parents captured 10–15 photographs of their caregiving responsibilities using smartphones or disposable cameras”. It's not clear how it can be 11 months of data collection if it took two weeks the photograph process.

Thank you very much for this point. We have mixed up the photo phase with the whole empirical phase of data collection and analysis. We have adjusted this information and wrote a statement at the end of point 3. Data analysis: The empirical process of data collection and analysis took place between January and November 2022. (L134)

  1. Please clarify the concept of “video group discussions”. Were they synchronous or asynchronous moments? Each participant presented all their 10-15 photos on that unique session? How was this sharing process?

Thank you for this question to make the process more transparent. We have added all these and more information in the data analysis paragraph (L107-109, 111-114).

  1. To better understand the data dimension, it would be important to have information on the session´s recording time and on the size of the transcripts.

Thank you and we agree. We have provided the information of the recording time (7.5 hours, L127) in “Familiarization with the data“. We think that information on size of the transcript is not very meaningful due to the computer-based analyses.

  1. Given the importance of this step to the process of conducting photovoice, it would be important to have more information regarding the additional meeting conducted with participants to review and confirm the results.

Thank you again for this very important input. We have reorganized and have rewritten the entire paragraph to ensure transparency. We believe this aspect represents a significant contribution not only to the findings but also to the participatory approach. (L159-167)

On the limitations of the study, the authors reflect on important features of the application of this methodology, but it could be enriched through the following aspects:

  1. What is the importance of the final exhibition according to Wang and Burris rationale?

The exhibition is often the final step of the Photovoice process. Although its impact is questionable, it is often criticized for not being implemented. The goal is to address politicians and decision-makers, who must also be present. Upon further consideration, I have decided that this is not "real" limitation. Therefore, we have decided to delete it.

  1. What the impact of a solo online format in the group dynamic. The group modality is so little considered that if this study had been in an individual format (interviews) it seems that it would be the same thing.

Thank you again for pointing this out. We are confident that this format was helpful and unique. In the data analysis paragraph, we added that the pictures shown by the participants sparked discussion, enabling the others to share their stories and provide context based on the photos. We don't think any of this would have been possible in a one-on-one interview setting.

Additionally, we believe it would be more challenging to avoid discussing a photo in a one-on-one setting due to the implicit pressure of fulfilling the "task." However, according to the participants, physical meetings are more effective in terms of opportunity for exchange. We noted this in the strengths and weaknesses paragraph. (L469)

  1. And for both questions, what future studies should consider and improve.

This is an excellent question—thank you for bringing it up. We believe that future research should focus on designing and evaluating comprehensive support systems that are tailored to the needs of aging parents with caregiving responsibilities and that promote the autonomy and well-being of their adult children. Additionally, we believe that the evaluation we conducted on photovoice should be carried out more systematically to ensure that participation is a meaningful process, not just a "label." We have reorganized the conclusion to center around these arguments. Thank you for prompting us to reflect on this further (L487-489 and L490-496).

  1. Once again, I congratulate the authors for such an important study.

Thank you once again for your appreciation.

Round 2

Reviewer 2 Report

Comments and Suggestions for Authors

The authors have satisfactorily addressed all previous comments and revision requests. The revised manuscript shows clear improvements in methodological clarity, data presentation, and the integration of relevant literature in the discussion section. In its current form, the study meets the journal’s scientific standards and is suitable for publication. I recommend acceptance.